# Association of the rs1966265 and rs351855 *FGFR4* Variants with Colorectal Cancer in a Mexican Population and Their Analysis In Silico

**DOI:** 10.3390/biomedicines12030602

**Published:** 2024-03-07

**Authors:** Irving Alejandro Carrillo-Dávila, Asbiel Felipe Garibaldi-Ríos, Luis E. Figuera, Belinda Claudia Gómez-Meda, Guillermo M. Zúñiga-González, Ana María Puebla-Pérez, Patricia Montserrat García-Verdín, Paola Beatriz Castro-García, Itzae Adonai Gutiérrez-Hurtado, Blanca Miriam Torres-Mendoza, Martha Patricia Gallegos-Arreola

**Affiliations:** 1División de Genética, Centro de Investigación Biomédica de Occidente (CIBO), Centro Médico Nacional de Occidente (CMNO), Instituto Mexicano del Seguro Social (IMSS), Guadalajara 44340, Jalisco, Mexico; irving.carrillo4754@alumnos.udg.mx (I.A.C.-D.); asbiel.garibaldi4757@alumnos.udg.mx (A.F.G.-R.); luisfiguera@yahoo.com (L.E.F.); monze_verdin@hotmail.com (P.M.G.-V.); 2Departamento de Biología Molecular y Genómica, Centro Universitario de Ciencias de la Salud (CUCS), Universidad de Guadalajara (UdeG), Guadalajara 44340, Jalisco, Mexico; beligomezmeda@gmail.com (B.C.G.-M.); itzgh02@hotmail.com (I.A.G.-H.); 3División de Medicina Molecular, Centro de Investigación Biomédica de Occidente (CIBO), Centro Médico Nacional de Occidente (CMNO), Instituto Mexicano del Seguro Social (IMSS), Guadalajara 44340, Jalisco, Mexico; mutagenesis95@hotmail.com; 4Laboratorio de Inmunofarmacología, Centro Universitario de Ciencias Exactas e Ingenierías (CUCEI), Universidad de Guadalajara (UdeG), Guadalajara 44430, Jalisco, Mexico; ampueblap@yahoo.com.mx (A.M.P.-P.); paola.castro@academicos.udg.mx (P.B.C.-G.); 5Laboratorio de Inmunodeficiencias Humanas y Retrovirus, División de Neurociencias, Centro de Investigación Biomédica de Occidente (CIBO), Centro Médico Nacional de Occidente (CMNO), Instituto Mexicano del Seguro Social (IMSS), Guadalajara 44340, Jalisco, Mexico; bltorres1@hotmail.com; 6Departamento de Disciplinas Filosófico Metodológicas, Centro Universitario de Ciencias de la Salud (CUCS), Universidad de Guadalajara (UdeG), Guadalajara 44340, Jalisco, Mexico

**Keywords:** colorectal neoplasms, *FGFR4* gene, variant, single nucleotide, in silico analysis, genetic variation, genetic predisposition to disease, pathogenies

## Abstract

The aim of this study was to associate *FGFR4* rs1966265 and rs351855 variants with colorectal cancer (CRC) in a Mexican population and to perform in silico analysis. Genomic DNA from 412 healthy individuals and 475 CRC patients was analyzed. In silico analysis was performed using the PolyPhen-V2, GEPIA, GTEx, and Cytoscape platforms. The GA genotype dominant model (GAAA) of rs1966265 and the AA genotype dominant and recessive models of rs351855 were identified as CRC risk factors (*p* < 0.05). CRC patients aged ≥ 50 years at diagnosis who consumed alcohol had a higher incidence of the rs351855 GA genotype than the control group (*p* < 0.05). Associations were observed between the rs1966265 GA genotype and patients with rectal cancer and stage III–IV disease. The rs351855 AA genotype was a risk factor for partial chemotherapy response, and the GA + AA genotype for age ≥ 50 years at diagnosis and rectal cancer was associated with a partial response to chemotherapy (*p* < 0.05). The AA haplotype was associated with increased susceptibility to CRC. In silico analysis indicated that the rs351855 variant is likely pathogenic (score = 0.998). Genotypic expression analysis in blood samples showed statistically significant differences (*p* < 0.05). *EFNA4*, *SLC3A2*, and *HNF1A* share signaling pathways with *FGFR4*. Therefore, rs1966265 and rs351855 may be potential CRC risk factors.

## 1. Introduction

Colorectal cancer (CRC) is a major clinical, social, and economic problem worldwide; it has the second highest incidence and is the third leading cause of death among all cancers. According to the latest estimates from GLOBOCAN, there were 1.9 million diagnosed cases of CRC and 900,000 deaths from CRC worldwide, with 14,901 cases of CRC and 7755 deaths due to CRC in Mexico [1,2].

Fibroblast growth factor receptors (FGFRs) are composed of a subfamily of receptor tyrosine kinases (RTKs), have high homology, and have been implicated in embryogenesis, angiogenesis, and tissue repair [3,4]. Given their nature and roles in various physiological pathways, it is not surprising that FGFR signaling has been implicated in several pathological conditions. It has been suggested that FGFRs mediate biological effects through four possible pathways: the RAS/MAPK, PI3-AKT, PLCγ, and JAK/STAT pathways [5].

The gene encoding FGFR4 (*FGFR4*) is located on chromosome 5q35.2 [6]. It spans approximately 11.3 kb [7] and has 18 exons ranging in size from 17 to 600 bp [8]. When *FGFR4* interacts with its ligands, it appears to activate the RAS/MAPK pathway; the resulting response is relatively weak but sufficient to promote proliferation [4,6]. *FGFR4* can also disrupt the cell membrane and subsequently cause changes in its composition, such as in actin [9]. One of the most studied *FGFR4* variants is the missense-type rs1966265 variant, which is located in exon 2 of the gene. This variant involves a change from adenine to guanine (A > G) and causes an amino acid substitution at codon 10 from isoleucine (Ile) to valine (Val) in the receptor signal peptide (p. Ile10Val) [10]. Although the literature mentions that a change to T can occur, there have been no reports to date on the possible allele frequencies of this change [11]. The NCBI has reported that the allele frequencies of this variant in Latin Americans of predominantly European and Native American ancestry are 0.69 for the reference allele (G) and 0.31 for the alternative allele (A).

Another variant, rs351855, has been identified that involves a sense change from guanine to adenine (G/A) in exon 9 (c.1162G > A) of the gene and results in an amino acid substitution of glycine for arginine at codon 388 of the protein (NC_000005.10:g.177093242G > A; p.Gly388Arg), which is within the transmembrane domain of the receptor [12,13,14]. This is one of the most common variants, with at least one copy identified in nearly 50% of the population [15]. This substitution has been associated with multiple genetic predispositions associated with poor outcomes and very poor overall survival [14,16]. However, its risk association with different types of cancer remains controversial [15]. The global allele frequencies reported for this variant by NCBI are 0.70 for the G allele and 0.30 for the A allele.

There are no current reports on the risk associations of these variants with CRC in Mexico, and global data are insufficient. Heinzle et al. [17] investigated the association between the rs351855 variant and CRC and found that in a subgroup of 71 patients who underwent surgery for CRC and expressed the variant, 20 patients exhibited *FGFR4* expression in the mucosa that was two times higher than normal levels. It has also been shown that rs351855 is correlated with the clinicopathological characteristics of the cancer but not with its appearance. In addition, Shiu et al. [10] found that the rs1966265 variant was associated with CRC metastasis.

In this study, we genotyped the rs1966265 and rs351855 variants, which are located in exons 2 and 9 of *FGFR4*, respectively, and performed an in silico analysis to determine the possible impact of the variants on *FGFR4* function. Numerous studies have shown that *FGFR4* has important functions in various signaling pathways [9,10,12,15,16,17]. Furthermore, these variants have been studied and associated with different types of cancer in different populations, but not in the Mexican population. Hence, it is necessary to determine whether there is an association between *FGFR4* variants and CRC in the Mexican population.

## 2. Materials and Methods

### 2.1. Experimental Subjects

A total of 5 mL of peripheral blood was collected from 412 healthy individuals and 475 clinically and histologically confirmed cases of CRC. 

The controls were obtained from healthy blood bank donors from the general population to ensure that they were the same as the cases. A questionnaire was completed by the donors that included age, sex, family history of hereditary cancer, personal pathological history, drug addiction, and tobacco and alcohol consumption, as well as a laboratory test that was normal.

Patients with CRC from the Hospital de Especialidades, CMNO were included. Demographic and clinical data were collected through a written questionnaire for this group. 

All participants gave written, informed consent. This study was conducted at the Centro de Investigación Biomédica de Occidente, Instituto Mexicano del Seguro Social, and was approved by the local ethics committee (CLIES #1305) as part of the project with the registration number R-2022-1305-081. All the procedures performed in the study were conducted under the 1964 Declaration of Helsinki, and written information was provided to the participants. Genomic DNA was extracted from the peripheral blood samples, quantified by nanodrops to obtain the DNA concentration, and then amplified by real-time PCR. 

### 2.2. Variant Analysis

Using TaqMan probes designed and validated by Applied Biosystems (Thermo Fisher Scientific, Waltham, MA, USA) and real-time PCR (qPCR), the rs1966265 and rs351855 samples were analyzed in a final volume of 10 μL and read using a CFX96 Real-Time PCR System (BIO-RAD, Berkeley, CA, USA) to genotype the variants by labeling with the above-mentioned fluorescent probes. As an internal control, 10% of the reactions were analyzed twice to ensure concordance across all analyzed samples.

### 2.3. In Silico Analysis

For the in silico analysis of the biological impact of the variants, the tool PolyPhen-2 (http://genetics.bwh.harvard.edu/pph2/ [accessed on 15 January 2024]) [18] was used to examine the changes at the protein level. PolyPhen-2 is used to analyze and calculate the probability that a given amino acid substitution is deleterious to a protein; it uses an algorithm that takes into account the physical aspects, structure, and function of the protein.

An in silico analysis of *FGFR4* expression was performed using colon adenocarcinoma (COAD) and rectal adenocarcinoma (READ) samples. The GEPIA tool (http://gepia.cancer-pku.cn [accessed on 15 January 2024]) [19], which integrates and analyzes data from The Cancer Genome Atlas (TCGA) and the Genotype-Tissue Expression (GTEx) repositories, was used. For this analysis, mean *FGFR4* expression data, previously normalized to the logarithmic scale log2 (TPM + 1), from 92 READ samples and 318 normal tissue samples were compared. Mean *FGFR4* expression data from 275 COAD tissue samples and 349 normal tissue samples were also compared. A significance level of *p* < 0.01 was set. A comparative analysis was performed to examine the mean *FGFR4* expression levels in the COAD and READ samples, segregated by tumor stage.

We also performed an eQTL analysis using the GTEx platform (https://www.gtexportal.org [accessed on 16 January 2024]) [20] to investigate the effect of each analyzed variant on *FGFR4* expression at the peripheral blood level in healthy donors.

To identify other proteins that may share signaling pathways or biological functions with *FGFR4* in the tumor environment, the GEPIA tool was used to detect genes with similar expression profiles to *FGFR4* in COAD and READ samples. Normal tissue samples obtained from the TCGA and GTEx repositories were included as controls. The top 20 genes with *FGFR4*-like expression in the selected samples were identified.

### 2.4. Analysis of the Relationships between Genes in Signaling Pathways

After identifying the top 20 genes with *FGFR4*-like expression, we investigated how these genes were related in terms of the signaling pathways of their protein products. We used the Cytoscape tool (https://cytoscape.org [accessed on 16 January 2024]) [21], which is widely used to analyze and visualize biological networks, to visualize interactions between previously identified genes and associated signaling pathways. Using this approach, we were able to explore how selected genes could interact in signaling cascades, share biological functions, and contribute to the pathophysiology of CRC. The genes found to interact with *FGFR4* in signaling pathways were analyzed using the cBIOPortal (https://www.cbioportal.org [accessed on 16 January 2024]) [22] to determine which are mutated in CRC and to what extent. We analyzed the expression profile data of 8615 samples from patients with CRC from different studies.

### 2.5. Statistical Analysis

The evaluation of the frequencies of the clinicopathological variables, genotypes, and alleles was conducted by analyzing the percentages obtained. The χ^2^ test was used to compare the observed and expected genotypes in the control group and to calculate the Hardy–Weinberg equilibrium (HWE). Statistical measures of odds ratios and binary logistic regression were used to analyze genotype associations using SPSS Statistic Base 24 software (Chicago, IL, USA). In addition, the online version of The SHEsis was used to analyze pairwise linkage disequilibrium (D′) and haplotype frequency [23]. Significance thresholds were set at *p* < 0.01 for the in silico analysis and *p* < 0.05 for genotyping, according to the recommendations of the software used.

## 3. Results

The demographic variables of the study groups are described in Table 1. The mean age in the group consisting of patients with CRC was 59.45 ± 12.34 years, and the mean age in the control group was 59.40 ± 13.37 years. In terms of gender distribution, 55% of the subjects in both groups were male. There were no significant differences in tobacco and alcohol consumption between the two groups (*p* > 0.05).

When the clinicopathological characteristics of the CRC group were analyzed, the most common characteristics were rectal cancer (51%), stage III cancer (37%), moderately differentiated histology (85%), and no lymph node metastasis (57%) (Table 2).

As shown in Table 3, the following characteristics were found to be CRC risk factors for the rs1966265 variant: GA heterozygous genotypes (odds ratio [OR] = 1.59, 95% confidence interval [CI] = 1.21–2.08, *p* = 0.001), the dominant model (OR = 1.92, 95% CI = 1.41–2.64, *p* = 0.001), the A allele (OR = 1.29, 95% CI = 1.07–1.57, *p* = 0.007), and the G allele (OR = 0.76, 95% CI = 0.63–0.93, *p* = 0.007). For the rs351855 variant, the following characteristics were found to be CRC risk factors: AA genotype (OR = 4.42, 95% CI = 2.77–7.06, *p* = 0.001), the dominant model (OR = 1.76, 95% CI = 1.34–2.30, *p* = 0.001), the recessive model (OR = 4.42, 95% CI = 2.77–7.06, *p* = 0.000), and the A allele (OR = 2.04, 95% CI = 1.65–2.52, *p* = 0.019) (Table 3).

A comparison of the data from CRC patients and the control group showed that the heterozygous GA genotype of the rs351855 variant was found to be a risk factor for those ≥50 years of age and those who consumed alcohol (Table 4). 

In the CRC group, the GA genotype of the rs1966265 variant was found to be associated with stage III–IV rectal cancer (OR = 1.83, 95% CI = 1.03–3.31, *p* = 0.044). The AA genotype of the rs351855 variant was associated with a partial response to chemotherapy (OR = 2.34, 95% CI = 1.02–5.36, *p* = 0.038). The dominant allele GA + AA (combined genotypes) was associated with an age at diagnosis of ≥50 years (OR = 1.94, 95% CI = 1.04–3.64, *p* = 0.038) and with rectal cancer and a partial chemotherapy response (OR = 3.2, 95% CI = 1.12–8.09, *p* = 0.028) (Table 5).

A comparison of the studied groups showed that there were statistically significant differences between the haplotype frequencies in the two groups: for GA, the OR = 14, 95% CI = 7.70–27.9, and *p* = 0.0001; for AA, the OR = 1.8, 95% CI = 1.45–2.42, and *p* = 0.0001; and for GG, the OR = 0.6, 95% CI = 0.50–0.75, and *p* = 0.0001 (Table 6). Regarding the linkage disequilibrium of the rs1966265 and rs351855 variants, *D*′ = 0.26 and r^2^ = 0.03 in the control group.

As shown in Figure 1, the frequencies of the A allele in the *FGFR4* rs1966265 and rs351855 variants in our control group were statistically different from the frequencies found in other global populations (*p* < 0.05). The exceptions for the rs1966265 variant were the Kinh population in Ho Chi Minh City, Vietnam (KHV), and the Chinese Dai population in Xishuangbanna, China (CDX). The exceptions for the rs351855 variant were the following populations: African-Caribbean in Barbados (ASW), Colombian from Medellin, Colombia (CLM), Utahns of Northern and Western European descent (CEU), Puerto Rican in Puerto Rico (PUR), Iberian populations in Spain (IBS), and Toscani in Italy (TSI). Allele frequency data for the other populations were obtained from the Ensembl platform (https://www.ensembl.org/Multi/Search/Results?q=rs1966265;site=ensembl_all and https://www.ensembl.org/Multi/Search/Results?q=rs351855;site=ensembl_all [accessed on 18 October 2023]).

### 3.1. In Silico Analysis of the Biological Impact of the Variants

The PolyPhen-V2 platform was used to analyze the biological impact of the variants, and the results showed that the rs351855 variant, which results in a Gly388Arg change, is “probably pathogenic”, with a score of 0.998, a sensitivity value of 0.27, and a specificity value of 0.99. The variant rs1966265, which results in an Ile10Val change, was classified as “benign”, with a score of 0.000 (Figure 2).

### 3.2. Analysis of the FGFR4 Expression Profiles in the Patient and Control Samples

The *FGFR4* expression profiles in the patient and control samples were analyzed using the GEPIA tool. Statistically significant differences (*p* < 0.001) were observed when comparing the mean *FGFR4* expression in the COAD samples (5.62) to that in the normal tissue samples (3.44) and when comparing the mean *FGFR4* expression in the READ samples (5.62) to that in the normal tissue samples (2.87) (Figure 3).

When the *FGFR4* expression profiles were stratified by tumor stage, no statistically significant differences were observed.

We then analyzed the genotypic expression of the *FGFR4* rs1966265 and rs351855 variants in peripheral blood samples taken from healthy donors. We used the GTEx database (https://gtexportal.org/home/testyourown [accessed on 15 January 2024]) and analyzed 670 blood samples. As shown in Figure 4, statistically significant differences (*p* < 0.05) were observed in the genotypic expression of the rs351855 variant.

### 3.3. In Silico Analysis of Genes with Expression Patterns Similar to FGFR4 and Their Relative Roles in Signaling Pathways

We performed a similar-gene analysis using the GEPIA tool and found that other genes had similar expression profiles to *FGFR4* in the READ and COAD tissue samples. Specifically, 20 genes were identified that had similar expression profiles to *FGFR4*: *RTKN*, *HNF1A*, *RNF43*, *RPN2*, *VDAC1*, *SLC1A5*, *ETV4*, *SAMD10*, *FANCF*, *ERG1C3*, *ASCL2*, *GPR35*, *SLC3A2*, *EFNA4*, *CBX8*, *GPX2*, *HM13*, *PIGU*, *CMTM8*, and *GYLTL1B* (Pearson’s correlation coefficient = 0.78–0.68). However, the analysis we performed using the Cytoscape tool showed that only EFNA4, SLC3A2, and HNF1A, which are involved in the phosphatidylinositol-3-kinase (PI3K)-AKT signaling pathway, shared signaling pathways with *FGFR4* (https://www.ndexbio.org/iquery/2d3c5c93-cd70-4cca-bb9e-e1682728b32a/enrichment/1b8d8526-5c6e-11ec-b3be-0ac135e8bacf, accessed on 15 January 2024). The expression profiles of the identified genes were subsequently analyzed using the cBioPortal platform (https://www.cbioportal.org/study/summary?id=coad_caseccc_2015%2Ccoad_cptac_2019%2Ccoad_silu_2022%2Ccoadread_dfci_2016%2Ccoadread_genentech%2Cbowel_colitis_msk_2022%2Ccoadread_tcga%2Ccoadread_tcga_pub%2Ccoadread_tcga_pan_can_atlas_2018%2Ccoadread_mskcc%2Ccoadread_mskresistance_2022%2Ccrc_apc_impact_2020%2Ccrc_dd_2022%2Ccrc_nigerian_2020%2Cmsk_met_2021%2Ccrc_msk_2017%2Crectal_msk_2019 [accessed on 15 January 2024]). The results showed that 2% of the analyzed samples contained *FGFR4* variants, 0.8% contained *EFNA4* variants, 3.2% contained *SLC3A2* variants, and 2.7% contained *HNF1A* variants. Among the patients found to have mutations in the above genes, 54% had COAD, 35.4% had colorectal adenocarcinoma, 7.8% had READ, and 2.8% had mucinous adenocarcinoma of the colon and rectum (Figure 5).

## 4. Discussion

Due to its high incidence and mortality rate, CRC is a significant health problem in Mexico and worldwide. CRC affects both men and women, and the average age at diagnosis is ≥50 years [25]. The mean age at diagnosis of the patients in this study was consistent with this previous finding. Regarding the type of cancer, we observed a slightly higher frequency of rectal cancer compared to colon cancer in our study group, as well as a higher incidence of stage III disease.

*FGFR4* is involved in important cell regulation and growth processes. Disruption of such processes is a hallmark of cancer and can result in cellular energy changes, proliferation, signaling, release of cellular immune factors, growth suppression, escape from immunity, angiogenesis, invasion, metastasis, genomic instability, and escape from programmed cell death [26,27].

It has been demonstrated that *FGFR4* is highly expressed in gastric cancer tissues, and this high expression is associated with a poor prognosis [6]. In a study of ovarian cancer, *FGFR4* was shown to be a prognostic indicator in advanced-stage disease, and inhibition of *FGFR4* with siRNA significantly inhibited ovarian tumor growth in vivo and in vitro [8]. In another study, hepatoma cell lines treated with FGF19 (a specific *FGFR4* agonist) showed reduced cell proliferation and invasion [28]. When the *FGFR4* inhibitor BLU9931 was tested in CRC cells, it was found to significantly inhibit cell proliferation and increase the rate of apoptosis in an inversely proportional manner [7]. Therefore, it has been suggested that *FGFR4* is an important target molecule for the treatment of CRC. 

It has been proposed that *FGFR4* is altered in various cancers [6,8,10,26,28], and the associations between the *FGFR4* rs1966265 and rs351855 variants and cancer have been described [29,30,31,32]. In a study conducted in a Chinese population, it was associated with the risk of developing breast cancer [29]. In a study of urothelial cell carcinoma, the rs1966265 variant was associated with tumor progression and histologic grade [30], and the rs351855 variant was associated with liver cirrhosis and increased AFP levels in hepatocarcinoma [31]. The rs351855 variant was also associated with metastasis in lung adenocarcinoma [32]. In irritable bowel syndrome-diarrhea, the rs1966265 and rs351855 variants have been associated with colonic transit [33,34].

In this study, we observed that the GA genotype, dominant model GA + AA, and A allele of the rs1966265 variant were associated with a risk of developing CRC. This is the first study to report on the association between rs1966265 and CRC susceptibility in a Mexican population. We also observed that the AG and AA genotypes, allele A, and dominant and recessive models of the rs351855 variant were associated with the risk of developing CRC.

We performed a comparative analysis to determine the A allelic frequencies of the rs1966265 and rs351855 variants in the control group and compare them with those in other global populations. The results highlighted the importance of conducting population studies and determining the allelic frequency in different populations because the frequencies in our population differed from those in the included European, East Asian, and African American populations. Our results suggest that ethnicity is a contributing factor to the observed variability in the frequencies of the *FGFR4* genotypes.

Heterozygous genotype (GA) of the rs351855 variant was found to be a risk factor in individuals aged ≥ 50 years who consumed alcohol when the CRC group was compared with the control group. No previous studies have identified or investigated this association. However, it has been reported that alcohol may be the key risk factor for neoplastic transformation [35].

Although cancers are detected in people of all ages, the highest frequency of detection is observed in individuals aged 50 to 60 years old, and the highest mortality rate is found in people aged ≥ 70 years old. Therefore, cancer is considered to be a disease associated with aging. It is known that aging is a biological process that overlaps with the characteristic features of cancer, which can cause genomic instability, increased inflammation, and senescence [36].

The results of our association analysis of the clinicopathologic variables of the CRC patients who carried *FGFR4* variants showed that being a carrier of the rs1966265 GA genotype was associated with rectal cancer and advanced-stage (stage III) disease. In the only related study reported in the literature, which included 443 patients with CRC and the same number of controls from the Chinese population, no association was observed between the A allele of the rs1966265 variant and progression in CRC patients [10].

In another study that was conducted among breast cancer patients and a control group from the Chinese population, the A allele was associated with the risk of breast cancer, and the GA and GG genotypes were associated with patients with lymph node-positive tumors [29]. Our results showed that in the Mexican population of our study, the frequency of the A allele of the rs1966265 variant is slightly higher than that of the G allele; therefore, the associations between the A allele and the clinicopathologic variables in the patients with CRC were remarkable.

It should be noted that the AA genotype was associated with a partial response to chemotherapy in colon cancer patients carrying the rs351855 variant, and the dominant GA + AA model was associated with an age ≥ 50 years and a partial response to chemotherapy in rectal cancer patients. Only one similar study has been conducted in Chinese patients with colon cancer; this found an inverse association (protection) between the AA genotype and CRC progression, which contrasts with our findings [10]. However, in a meta-analysis evaluating the association between the rs351855 (AA) variant and the prognosis of several types of cancer (according to nodal status and overall survival), a higher risk of poor overall survival was found compared to homozygous carriers of the common GG genotype, even after adjusting for nodal status [37].

Another study performed on cell lines with different endogenous expression patterns to overexpress either the *FGFR4* GG or *FGFR4* AA genotypes revealed the biological importance of both variants in cell growth and migration. In the same study, it was found that patients with CRC who carried the AA genotype (*n* = 182) were found to have a fivefold higher risk of having tumors that were stage II or greater [17]. Although no other studies have been conducted to support these findings, these results indicate that the rs1966265 and rs351855 variants may affect different clinicopathologic features, and thus it is important to stratify by these characteristics to understand the associations between the variants and CRC risk. 

The AA genotype of the rs351855 variant, which encodes an arginine at codon 388 of the transmembrane domain of FGFR4, has been shown to increase cancer risk. When FGFR4 is transphosphorylated, it is activated and induces cell proliferation by stimulating the kinase receptor [38]. The presence of this arginine stabilizes the protein and prolongs its activity [39].

It is known that the expression of FOXC1 is increased in metastatic CRC cells when it is bound to integrin α7 (ITGA7) and FGFR4 [33]. It has also been shown that there are inhibitors that block the FGFR4-FRS2-ERK signaling pathway and restrict cancer cells that exhibit glycolytic phenotypes and chemoresistance [40]. Ahmed et al. found that inhibition of FGFR4 can attenuate RAD51-mediated double-stress break (DSB) repair, thereby attenuating the anti-radiation effect in CRC cells [41].

Heinzle et al. reported that although the variant may not play a role in CRC initiation, it may influence the significantly accelerated progression of the disease [17]. This substitution has been associated with poor outcomes and very low overall survival in individuals who carry it, regardless of ethnicity [17]. It has also been reported that signaling mediated by the signal transducer and activator of transcription-activated pathways 3 (STAT3) makes a major contribution to the increased risk of cancer and tumor progression; when the signaling pathway is activated, it causes a deregulation of cell proliferation and apoptosis [13].

Most likely, the imbalance of *FGFR4* and cancer cell survival by the variant alleles of rs1966265 (A) and rs351855 (A) is because they are located at the target sites of cellular recognition and modulation [38].

The *FGFR4* variants rs1966265 and rs351855 analyzed in this study were not in linkage disequilibrium and showed a non-significant association. The GA and AA genotypes, present in 9% of the controls and 14% of the patients with CRC, and in 14% of the controls and 24% of the patients with CRC, respectively, were observed as possible risk factors for CRC susceptibility. A study of patients with oral squamous cell carcinoma showed that individuals with the AA genotype had a higher risk of developing oral squamous cell carcinoma than individuals without the AA genotype [42]. It should be noted that further population studies are needed to confirm this association.

The in silico analysis provided valuable information for the identification of genes and metabolic pathways involved in the cellular modulation of CRC development and its variability. Using the PolyPhen-V2 platform [18], the rs351855 variant was predicted to be probably pathogenic, with a score close to 1. Using the GEPIA tool [19], we compared the *FGFR4* expression levels in colon and rectal cancer tissue samples with those in normal tissue samples. We also examined the expression of the variants in peripheral blood samples from healthy donors. Using the Cytoscape platform [21], we showed that only EFNA 4, SLC3A2, and HNF1A, which are involved in the PI3K-AKT signaling pathway, share signaling pathways with FGFR4.

Complementary analyses revealed that the rs351855 variant influenced *FGFR4* expression in various human tissues, and increased FGFR4 levels were correlated with the presence, advanced stage, and distal spread of COAD using data from GTEx [20]. These findings suggest that amino acid changes and modified expression of *FGFR4*, attributable to the genetic variant, may influence the progression of CRC.

Members of the FGFR family function as tyrosine kinase receptors, and five types are known that differ by the presence (FGFR14) or absence (FGFR5 or FGFRL1) of the intracellular kinase. In FGF–FGFR binding, the receptor and ligand form a dimer that stimulates the autophosphorylation complex and important signaling pathways that regulate cell growth, such as AKT (serine-threonine protein kinase), MAPK (activated protein kinase), mitogens, and STAT3. FGF19- and FGFR4-mediated activation of the PI3K-AKT, MAPK, STAT3, and epithelial-mesenchymal transition (EMT) pathways may be involved in malignancy [43]. FGFR is known to be an important factor in tumor cell production, angiogenesis, migration, differentiation, aggressiveness, and drug resistance. It plays a fundamental role in cancer by participating in signaling and tumor progression and presenting some small molecular inhibitors that target FGFR4. FGFR4 can alter the production of relative transcription factors directly related to patient survival. FGFR4 binds to FGF19, an important factor in the development and progression of several cancers [44].

In a genetic epistasis analysis, it was observed that when FOXC1 binds to ITGA7 and FGFR4, its expression is increased in metastatic CRC cells, and therefore FGFR4 has been proposed as a potential drug target [40]. It has also been shown that the MAPK/ERK signaling pathway is activated by FGFR4 and FGF receptor substrate 2 (FRS2) phosphorylation in drug-resistant cells and that inhibitors that block the FGFR4-FRS2-ERK signaling pathway limit glycolytic phenotypes and chemoresistance [36]. Ahmed et al. investigated how CRC cells resist radiotherapy through the expression of FGFR4 and observed that inhibition of FGFR4 can attenuate RAD51-mediated DSB repair, thereby attenuating the anti-radiation effect [41]. FGF19 overexpression also plays a critical role in drug resistance by promoting EMT to activate the GSK3β/β-catenin and STAT3 signaling pathways [45,46]. The FGF19/FGFR4 signaling pathway is involved in the metabolism and maintenance of cell growth. Suppression of the expression of FGFR4 and its ligand or impairment of their subsequent activation is considered to be a major cause of tumor growth [45,46]. FGFR4, like three other FGFRs, has three immunoglobulin-like domains (IgI, IgII, and IgIII) that are required to bind a specific ligand. More importantly, unlike FGFR1–3, there are no IgIII splice variants in FGFR4, suggesting that the development of selective FGFR4 inhibitors may be an effective therapeutic strategy [47].

A significant association of FGFR4 expression with the FGF19 ligand has been observed in cancer progression and metastasis, making it a promising therapeutic target against cancer [45]. It is also mentioned that FGFR4 is a target for patients with breast cancer due to the co-expression of FGFR4 and FGF19, associating it with the expression of phosphorylated AKT. On the other hand, the target objective focuses on reducing the expression levels of FGFR4 and FGF19. Zaid et al. observed that when the expression levels of FGFR4 and FGF19 are decreased, it significantly reduces tumor growth in vivo and in vitro in ovarian cancer [48] and in the cell line of colon cancer. The depletion of FGFR4 suppresses the proliferation and migration of colon cancer cells. However, one of the major obstacles to cancer therapies is the drug resistance that patients can develop [17]. An increase in FGFR4 expression has been observed in doxorubicin-resistant breast cancer patients. In addition, silencing or inhibiting FGFR4 in patients with CRC has been observed to make cancer cells more sensitive to two types of treatments against CRC: 5-FU (5-fluorouracil) and oxaliplatin [49]. There is evidence that inhibition of the caspase-3 pathway reduces ROS-induced apoptosis in colon epithelial cells [45]. *FGFR4* Arg388 was also found to decrease proliferation, invasion, and metastasis in prostate cell lines [49]. Another study showed that in patients with neuroblastoma, the AA genotype and the A allele of the Gly/Arg388 variant were more frequent than in controls, demonstrating that regulation of FGFR4 has slower degradation than the Gly388 receptor in neuroblastoma cells and reduced internalization into multi-vesicular bodies [50]. On the other hand, in breast cancer patients, the minor alleles of rs1966265 and rs351855 in FGFR4 were shown to be strongly associated with breast cancer, and the variant in turn was associated with lymph-node positivity [29]. Thus demonstrating the importance of FGFR4 as a pharmacological treatment for the inhibition of cancer in both in vitro and in vivo studies and as an inhibitor of tumor proliferation and invasion [17,39,40,41,42,43,44,45,46,47,48,49,50].

The identification of genes with expression profiles similar to *FGFR4* in CRC opens new lines of research in CRC and other pathologies. It should be noted that only *EFNA4*, *SLC3A2*, and *HNF1A* are involved in the same signaling pathways as FGFR4. The common MAPK pathway also involves EFGR, RET/PTC, KRAS, C-JUN, and AP1/SP1, and the common PI3-AKT pathway involves ET1 and PTEN. Thus, the genes that encode these proteins may be important factors in the development of the tumor microenvironment. It is known that tumorigenesis involves multiple genetic alterations, mainly in oncogenes and tumor suppressor genes, in normal cells that cause them to transform into neoplastic cells. These genetic changes result in the creation of the tumor microenvironment, which consists of elements that support the growth and survival of tumor cells and contribute to resistance to invasion by unsupportive cells. 

The *EFNA4* gene encodes the protein epinephrine 4, which belongs to the epinephrine family of transmembrane proteins. Together with their receptors, these proteins act in the nervous system and during erythropoiesis, as well as in the proliferation and metastasis of various types of cancer through different signaling pathways [51]. The SLC3A2 (solute carrier family 3 member 2) gene encodes a transporter protein that is involved in the regulation of intracellular calcium levels and transports amino acids (e.g., leucine and glutamine) that are essential for metabolic activation and cellular function as facilitators of plasma membrane transport. It has been linked to the mTORC1 pathway, a key metabolic regulator that stimulates cellular metabolism and activates the expression of c-Myc, a transcription factor for cell growth and proliferation. Certain metabolic and amino acid transport elements, such as SLC1A5 and SLC7A5, which are required for glutamine uptake, and the SLC3A2 subunit, which is required for leucine uptake, may influence the ability of T and CAR-NK cells to effectively fight tumors [52]. The *HNF1A* gene encodes hepatocyte nuclear factor-1 alpha (HNF-1α), which is important in the pancreas and liver. It acts as a transcription factor to regulate insulin, controls genes that regulate cell growth and survival, and acts as a tumor suppressor [53]. 

It should be noted that one of the limitations of this study is that it does not offer a more comprehensive analysis of the environmental and lifestyle factors along with the genetic data obtained, which could provide a more integral view of CRC risk. Studies are needed to understand these risk factors.

## 5. Conclusions

In this study, the rs1966265 and rs351855 variants were associated with the risk of developing CRC. People with CRC who were aged ≥ 50 years at diagnosis and who consumed alcohol had a significantly higher incidence of the rs351855 GA genotype than those in the control group. Among patients with CRC, the GA genotype of the rs1966265 variant was also associated with rectal cancer and advanced-stage disease (stages III–IV). The AA genotype of the rs351855 variant was also found to be a risk factor for partial response to chemotherapy, and the GA + AA genotype was found to be associated with age ≥ 50 years at the time of diagnosis and rectal cancer with a partial response to chemotherapy. The AA haplotype is found to be a risk factor for CRC susceptibility. The in silico analysis showed that the rs351855 variant was likely to be pathogenic. When *FGFR4* expression was analyzed in blood samples, the rs351855 genotypes were found to be expressed at significantly different levels. *EFNA4*, * SLC3A2*, and *HNF1A* were found to have similar expression profiles to *FGFR4* and to be involved in the PI3K-AKT signaling pathway. Further studies are needed to confirm the observed results.

## Figures and Tables

**Figure 1 biomedicines-12-00602-f001:**
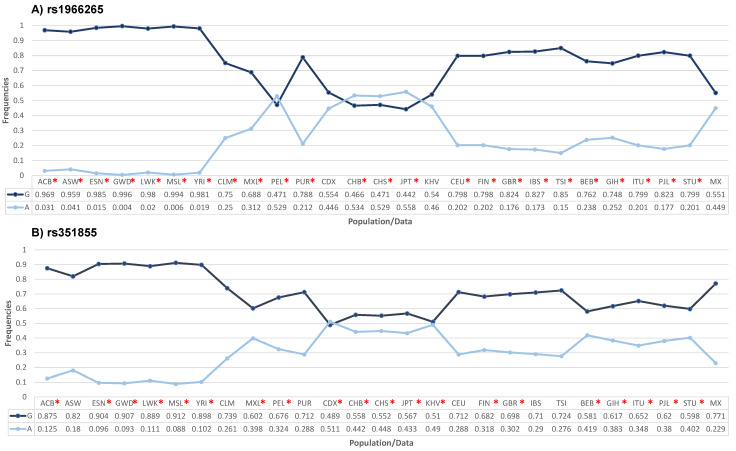
Comparison of the allelic frequencies of the *FGFR4* rs1966265 (**A**) and rs351855 (**B**) variants in different populations with the frequencies found in the Mexican population used in this study. * *p* < 0.05. ACB: African Caribbean in Barbados; ASW: African Ancestryin Southwest US; ESN: Esan in Nigeria; GWD: Gambian in Western Division; LWK: Luhya in Webuye, Kenya; MSL: Mende in Sierra Leone; YRI: Yoruba in Ibadan, Nigeria; CLM: Colombian in Medellin; MXL: Mexican ancestry in Los Angeles, California; PEL: Peruvian in Lima; PUR: Puerto Rican in Puerto Rico; CDX: Chinese Dai in Xishuangbanna; CHB: Han Chinese in Beijing; CHS: Southern Han Chinese; JPT: Japanese in Tokyo; KHV: Kinh in Ho Chi Minh City, Vietnam; CEU: Utah residents with Northern and Western European ancestry; FIN: Finns in Finland; GBR: British in England and Scotland; IBS: Iberian populations in Spain; TSI: Toscani in Italy; BEB: Bengali in Bangladesh; GIH: Gujarati Indians in Houston; ITU: Indian Telugu in the UK; PJL: Punjarbi in Lahore, Pakistan; STU: Sri Lankan Tamil in the UK; and MX: Mexican population analyzed in the present study. The frequencies of the other populations were obtained from https://www.ensembl.org/Multi/Search/Results?q=rs8720;site=ensembl_all and https://www.ensembl.org/Multi/Search/Results?q=rs12587;site=ensembl_all [accessed on 18 October 2023]).

**Figure 2 biomedicines-12-00602-f002:**
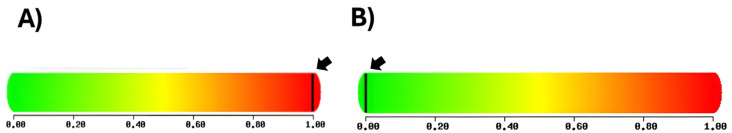
(**A**) The rs351855 variant was predicted to be pathogenic, with a score of 0.998, a sensitivity value of 0.27, and a specificity value of 0.99. (**B**) The rs1966265 variant was predicted to be benign, with a score of 0, a sensitivity value of 1.0, and a specificity value of 0.0 (http://genetics.bwh.harvard.edu/pph2/ [accessed on 16 January 2024]) [18].

**Figure 3 biomedicines-12-00602-f003:**
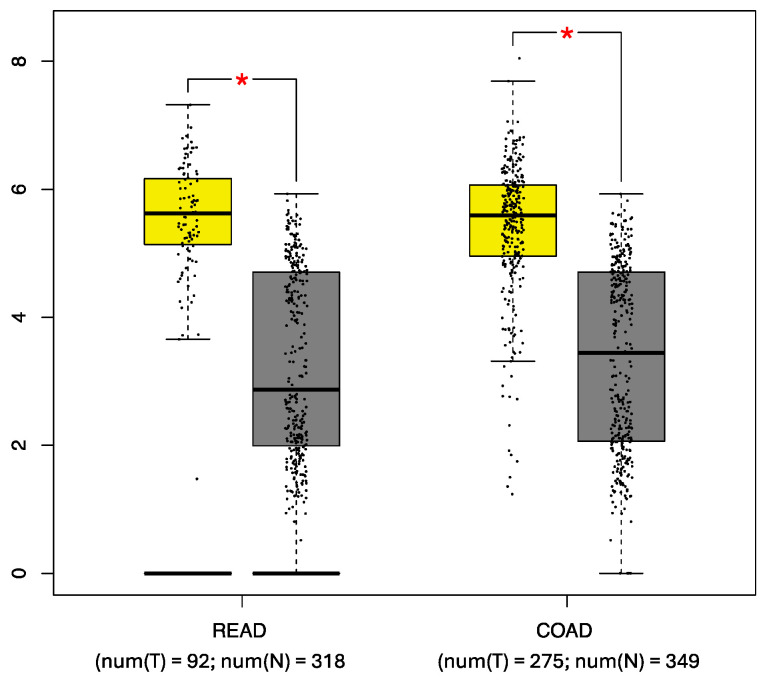
Comparative analysis of *FGFR4* expression. The expression in rectal adenocarcinoma (READ) and colon adenocarcinoma (COAD) samples (yellow box) was compared with that in control tissue samples (gray box). num(T) Number of tumor tissue, and num(N) Number of normal tissue, * *p* < 0.05. (http://gepia.cancer-pku.cn/ [accessed on 15 January 2024]) [19].

**Figure 4 biomedicines-12-00602-f004:**
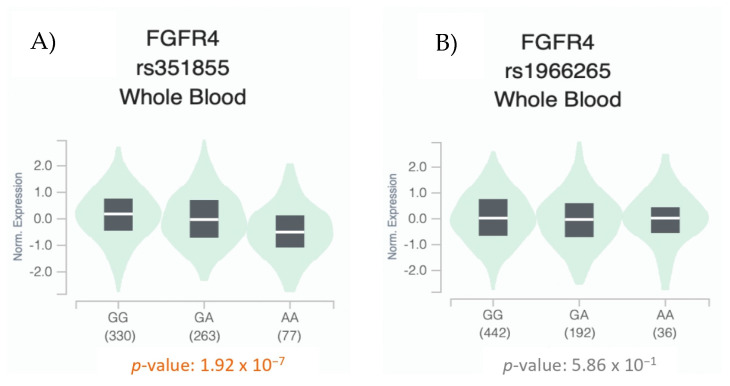
Whole blood eQTL analysis of the *FGFR4* rs351855 (**A**) and rs1966265 (**B**) variants according to the distribution of their genotypes. The width of the violin plot indicates the density of the data (data were obtained from GTEx (https://gtexportal.org/home/testyourown [accessed on 15 January 2024]) [20].

**Figure 5 biomedicines-12-00602-f005:**
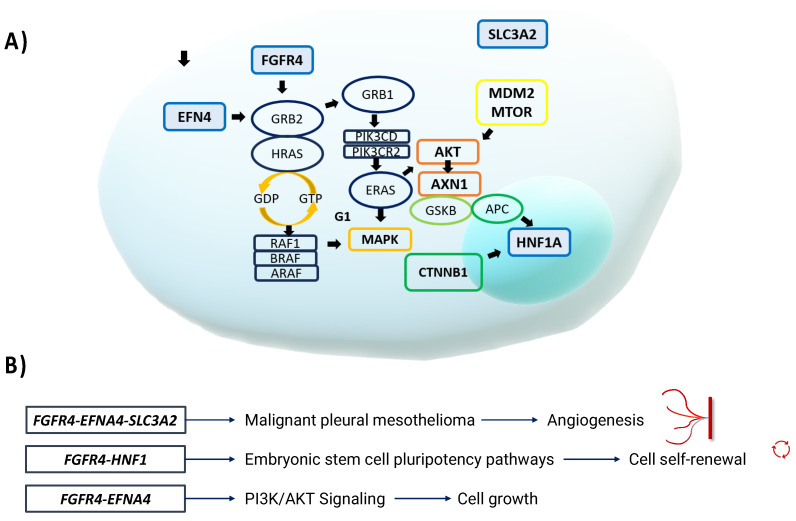
(**A**) *EFNA4*, *FGFR4*, *SLC3A2*, and *HNF1A* are involved in signaling pathways that may promote carcinogenesis. Within these pathways, RAS proteins, PI3K, AKT, mTOR, and other proteins promote cell proliferation, carcinogenesis, survival, and various carcinogenic events. (**B**) *FGFR4* and *HNF1* are involved in the pluripotency pathways of embryonic stem cells. For example, the TGF-β signaling pathway transmits signals to Smad2/3/4 and FGFR, which in turn activate the MAPK and AKT pathways to promote cell self-renewal [24]. *FGFR4*, *EFNA4*, and *SLC3A2* are also involved in different pathways involved in the development of malignant pleural mesothelioma, where *RAS* family genes *TP53*, *AKT*, and *APC* also participate in promoting angiogenesis. *FGFR4* and *EFNA4* together participate in the PI3K-AKT signaling pathway, which promotes cell survival and growth.

**Table 1 biomedicines-12-00602-t001:** The demographic data of the studied groups.

	CRC Patients (*n* = 475)		Controls (*n* = 412)		*p*
**Age at diagnosis (years)**					
Mean (SD) *	59.45	(12.34)	59.40	(13.37)	0.951
≤49 (*n*, %) **	92	19	81	20	1.0
≥50 (*n*, %) **	383	81	331	80	
**Sex**					
Male (*n*, %) **	260	55	225	55	1.0
Female (*n*, %) **	215	45	187	45	
**Tobacco consumption**					
Yes (*n*, %) **	161	34	142	34	0.94
No (*n*, %) **	314	66	270	66	
**Alcohol consumption**					
Yes (*n*, %) **	178	37	145	35	0.50
No (*n*, %) **	297	63	267	65	

SD (standard deviation); * Student’s *t*-test; ** χ^2^ test.

**Table 2 biomedicines-12-00602-t002:** General description of clinicopathological characteristics of the CRC group.

CRC Patients (*n* = 475)					
	** *n* **	**%**		** *n* **	**%**
**Location**			**Histological classification of adenocarcinoma**		
Rectum	242	51	Moderately differentiated	404	85
Colon	233	49	Not differentiated	27	5
**Stage**			Differentiated	44	10
I	16	3			
II	154	32	**Lymph node metastasis**		
III	177	37	Positive	203	43
IV	128	28	Negative	272	57

**Table 3 biomedicines-12-00602-t003:** Genotypic and allelic frequencies of the *FGFR4* rs1966265 and rs351855 variants in the studied groups.

Variant			CRC Patients		Controls *		OR	95% CI	*p*
**rs1966265**	**Model**	**Genotype**	***n* = 455**	**%**	***n* = 392**	**%**			
		GG	88	19	124	32	1.0		
		GA	266	58	184	47	1.59	1.21–2.08	0.001
		AA	101	23	84	21	1.04	0.75–1.45	0.851
	Dominant	GG	88	19	124	32			
		GA + AA	367	81	268	68	1.92	1.41–2.64	0.001
	Recessive	AA	101	23	84	21	1.04	0.75–1.45	0.787
		GG + GA	354	77	308	79			
		**Allele**	2*n* = 910		2*n* = 784				
		G	442	49	432	55	0.76	0.63–0.93	0.007
		A	468	51	352	45	1.29	1.07–1.57	0.007
**rs351855**	**Model**	**Genotype**	*n* = 460	%	*n* = 406	%			
		GG	212	46	244	60	1.0		
		GA	148	32	138	34	1.07	0.80–1.42	0.685
		AA	100	22	24	6	4.42	2.77–7.06	0.001
	Dominant	GG	212	46	244	60			
		GA + AA	248	54	162	40	1.76	1.34–2.30	0.001
	Recessive	AA	100	22	24	23	4.42	2.77–7.06	0.001
		GG + GA	360	78	382	77			
		**Allele**	2*n* = 920		2*n* = 812				
		G	572	62	626	77	0.48	0.39–0.60	0.019
		A	348	38	186	23	2.04	1.65–2.52	0.019

* Hardy–Weinberg equilibrium (HWE) values for the control group: rs1966265: χ^2^ test = 1.03, *p* = 0.309; and rs351855: χ^2^ test = 0.57, *p* = 0.44.

**Table 4 biomedicines-12-00602-t004:** Associations between the *FGFR4* rs351855 variant and two variables in the CRC patient group compared with the control group.

Variant	Genotype	Variable	OR	95% CI	*p*
rs351855	GA	≥50 years old	1.59	0.45–0.13	0.001
	GA	Alcohol consumption	1.58	0.46–0.13	0.001

Bivariate analysis, significance: *p* < 0.05.

**Table 5 biomedicines-12-00602-t005:** Associations between the *FGFR4* variants (rs1966265 and rs351855) and clinicopathological variables in the CRC group.

Variant	Genotype	Variable	OR	95% CI	*p*
rs1966265	GA	Rectal cancer and III–IV stage	1.83	1.03–3.31	0.044
rs351855	AA	Partial chemotherapy response	2.34	1.02–5.36	0.038
	GA + AA	≥50 years old	1.94	1.04–3.64	0.038
	GA + AA	Rectal cancer and partial chemotherapy response	3.2	1.12–8.09	0.028

Bivariate analysis, significance: *p* < 0.05.

**Table 6 biomedicines-12-00602-t006:** Haplotype frequencies of the *FGFR4* variants (rs1966265 and rs351855) in the studied groups.

	CRC Group 2*n* = 870			Control Group 2*n* = 772			
rs1966265	rs351855	*n*	%	*n*	%	OR (95% CI)	*p*
G	A	124	14	72	9	14 (7.70–27.9)	0.0001
A	A	207	24	110	14	1.8 (1.45–2.42)	0.0001
G	G	302	35	356	46	0.6 (0.50–0.75)	0.0001
A	G	237	27	234	31	0.8 (0.69–1.06)	0.187

## Data Availability

Data and materials are available in the article.

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
