# Peer review of "Association of the rs1966265 and rs351855 FGFR4 Variants with Colorectal Cancer in a Mexican Population and Their Analysis In Silico"

_biomedicines, 2024, doi:10.3390/biomedicines12030602_

Round 1

Reviewer 1 Report

Comments and Suggestions for Authors

I understand the rarety of pieces of information regarding the proposed genetic mutation. For readers try to link to classical biomarkers, prognostic factors, or standard treatments available (you mentioned some in the text ). Make it more clear. 

Thank you

Reviewer 2 Report

Comments and Suggestions for Authors

The authors present a study analyzing the association of the rs1966265 and rs351855 FGFR4 variants with colorectal cancer in a Mexican population, including 479 colorectal cancer patients and 412 healthy individuals.

The following issues need to be addressed:

1. The analysis involved two well-known and previously studied genetic variants in gene FGFR4 (rs1966265 and rs351855). Although the in silico analysis adds to our current knowledge and to the results of the study, the purpose of using mentioned silico tools should be described in the Introduction section. Similarly, in the Discussion section, the authors should add and discuss previous findings on the functional characterization of the studied variants. Although functional data are limited, there a few, e.g. PMID: 26840079, PMID: 29603419, PMID: 21622724.

2. Table 3 contains several errors; please correct. E.g. wrong percentages for Recessive AA and GG+GA!; wrong data format for allele percentages; correct 2.7 to 2.77 in rs351855 analysis for AA,… Please also check if there are any other errors.

3. Correct lines 175 and 176 p =0.000 to p = 0.001 as in Table 3. Line 175 correct also 2.7 to 2.77.

4. In the Statistical analysis, the authors stated that “Significance thresholds of p < 0.01 and p < 0.05 were set.” Please explain why there are two different thresholds, and in which cases 0.05 and in which a stricter 0.01 was used.

5. In the MM section, Variant analysis (lines 108-110), correct to the fitting symbol for micro (10 µL). What was the purpose of using a PCR cycler C1000?

6. In the Results section (lines 194-196) please explain what “dominant allele GAAA” stand for? Combined genotypes? Similar also stands for data presented in Table 5. Please add an explanation.

7. Correct Figure 2; mark A and B clearly, and move the "mark point" for rs351855 to the right position!

8. Please check throughout the entire manuscript for other possible errors, e.g. gene name in italics, …

Round 2

Reviewer 2 Report

Comments and Suggestions for Authors

The authors have addressed all the questions, and the manuscript is improved and almost suitable for publication.

I have three minor comments:

1. I don’t think the authors used the “C1000 Touch Thermal Cycler” as described in the MM section. Hence, my suggestion is to remove it. Genotypes were probably determined just with “CFX96 Real- Time PCR System (BIO-RAD, Berkeley, CA).”

2. Please explain the difference in the numbers of CRC patients in Table 1 (479 patients) and Table 2 (475 patients).

3. Table 3 still contains inconsistencies; there is a wrong data format for allele percentages (0.4857 should be corrected to 49, etc.).

Author Response

Dear reviewer,

           We are pleased to inform you that we have taken into account the latest comments received on our work and have made the appropriate corrections.

           Concerning the first comment, we have corrected the error noted and have specified in the materials and methods section that genotyping was carried out using the "CFX96 Real-Time PCR System".

           Regarding the second comment, we regret the confusion and have adjusted the number of CRC patients to 475, as is correct. Additionally, we have modified the corresponding table to accurately reflect these changes.

           In response to the third comment, we have corrected the inconsistencies in the table and have adjusted the frequencies, thus facilitating the interpretation of the percentages.

           We thank you again for your comments and suggestions, which have contributed to improving the quality of our work.

Best regards, 

Dr. Martha Patricia Gallegos Arreola